# Public sector's efficiency as a reflection of governance quality, an European Union study

**Camelia Negri, Gheorghiţa Dincă**\*

Department of Finance and Accounting, Transilvania University of Braşov, Braşov, Romania

\* gheorghita.dinca@unitbv.ro

**Data Availability Statement:** All relevant data are within the paper and its Supporting Information files.

**Funding:** The authors received no specific funding for this work.

## Abstract

The main objective of this paper is to assess the efficiency of the European Union's public sector from a quality of governance approach, employing a two-step methodology. In the first stage, EU states' efficiency scores are determined using Data Envelopment Analysis. Once quantifying public efficiency, the second step of the methodology examines the determinants of efficiency using a quantile regression estimation technique, with an emphasis on demographic features, corruption, economic freedom, and governmental digitalization. The novelty is provided by the assessment of the public sector's efficiency and efficacy from a broader approach in order to determine the efficiency scores of European Union countries, as well as the key factors that may impact the public sector's performance. The main findings, namely that governance quality can be considered an important resource in analysing public performance and that human resources, freedom, democracy, corruption, and digitalization have an impact on efficiency, are important considerations not only for policymakers but also for society, researchers, and the academic community. Reform measures should strive to improve both the technical and democratic components of public institutions in order to more effectively and transparently allocate public resources, while taking into consideration local and national peculiarities.

## Introduction

In a time of most unexpected changes and challenges (global economic and financial crisis, resource crisis, pandemic crisis, demographic changes, migration, climate change, technological progress, etc.), good governance and efficiency of the public sector are topics that require in-depth analysis by scholars, politicians, officials, and entrepreneurs to identify the best policies and practices and to take a step forward and implement them where reforms are required or where the need to improve the performance of public institutions is mandatory. A reality today is the growing importance of the public sector in coping with change. Thus, states need to reinvent and modernise themselves and become more efficient to better fulfil their social functions. In addition, states are compelled to intervene to protect markets by fostering and supporting initiatives and procedures that require public officials and managers to conduct innovative public sector measures [1].

The literature presents many approaches to the concepts of governance and good governance as well as their impact on economic development, but less on the link between good

**Competing interests:** The authors have declared that no competing interests exist.

governance and public sector performance, efficiency and the factors that may influence this efficiency.

Thus, the current paper aims at filling this specific gap by thoroughly examination of this relationship, starting from the following questions:

1. How important is the quality of governance in assessing the efficiency of the public sector as a whole?

2. What are the drivers of the public sector's efficiency, with a focus on human resources and economic freedom, democracy, corruption, and digitalization?

The main purpose of the article is to assess the efficiency of the European Union's public sector from a quality of governance approach, the novelty lying on emphasizing governance an indispensable resource in achieving high levels of public performance.

Until reaching the final conclusions and remarks of the paper regarding the link between governance quality, public efficiency and the determinant factors of efficiency score, the paper aims at the following intermediate objectives:

- Review of the main ideas and theoretical benchmarks for governance quality and public sector efficiency;

- Application of the research methodology in two stages:

○ Establishing the inputs and outputs for the DEA methodology, applying DEA to obtain efficiency scores for the EU states, followed by the interpretation of the results;

○ Building a regression model in which the dependent variable is the efficiency score obtained in the first stage and the independent variables target the areas of human resource quality and population characteristics, economic freedom and democracy, corruption, and digitalization.

- Analysis of the results and correlations obtained and their economic implications;

- Policy implications.

Thus, the study's main assumptions are that firstly, governance plays an important role in boosting public performance in the case of EU structure, countries recording high scores of technical and democratic governance performing better than those who record lower scores and secondly, that human and population characteristics, together with the levels of democracy, corruption and digitalization significantly influence these obtained efficiency scores.

To achieve these goals and to test the aforementioned hypotheses, the methodology is divided into two main parts, the same as the methodological framework used by Moutinho et al. [2] in their study: performing a Data Envelopment Analysis (DEA) to determine the public efficiency for the EU27 countries, thus quantifying the concept of public performance, followed by a quantile regression to further explain the efficiency scores obtained, using independent variables that portray population characteristics, as well as levels of perceived corruption, economic freedom, and government digitalization

This paper presents a complex approach regarding the quality of governance and the efficiency and efficacy of the public sector. The findings are relevant not only to the European Union structure but also to other regions or countries. The main results are important for policymakers as well as for society, scholars, and the academic community. The scientific results show the significant relationship between the quality of governance and public sector efficiency, and also fill the gap in this research area by further investigating the exogenous determinants of public efficiency, considering the distribution of public efficiency scores.

## Literature review

Governance refers to the ability and capacity of state institutions to serve their own citizens. This involves a specific set of intertwined rules, processes, and attitudes which allow governments to enforce their power into societies. Thus, governance is found to be a base indicator for the stability and performance of a society [3]. The majority of studies focus either on a national or cross-national analysis of governance quality, corruption levels and rule of law [4–11].

More recently, the above-mentioned topics of interest have received a multi-layered approach, respectively at a sub-national and regional level inside the EU. This research strategy puts forward substantial differences between regions and emphasises the significance of governance quality and its effects [12, 13]. The authors highlight that regional discrepancies, in terms of economic growth, productivity, and employability have deepened, contrary to the EU's target of convergence. The quality of public institutions is the key factor in maintaining sustainable economic and social development [13–16].

Governance quality is directly connected with public sector's performance, quality, and quantity of public services, an efficient allocation of public resources, and ultimately follows to achieve citizens' satisfaction through increasing living standards [1, 17–20].

Governments are currently facing complex challenges in reducing poverty and social inequalities, addressing the energy crisis, food shortages, a growing migration phenomenon, climate change, and to some extent, increasing military costs. To address these challenges, governments have to increase their efficiency in managing public budgets and even identify new revenue streams. Moreover, the ongoing effort of improving public institutions and governance quality is also motivated by public expectations, which occupy an incontestable role in this process.

The concept of efficiency describes and measures how effectively and rationally resources are utilised by an organisation to achieve its specific goals [21]. Furthermore, efficiency is the capacity to perform the proper actions with the least amount of resources and the maximum results. To measure the public sector's efficiency and performance, Afonso et al. [22] developed a system of indicators that includes the basic functions defined by Musgrave (allocation, distribution, and stabilization) and several specific indicators that promote equal opportunities in the market. In addition, Afonso et al. [22] established these indicators (in correlation with the costs of achieving them) to measure public sector's performance for 23 industrialised countries, the results showing that economies with limited and narrow public departments have higher economic performance, while economies with larger public sectors and organisations have a more evenly spread income distribution. Continuing this line of research, Afonso et al. [23] have constructed an aggregate indicator with the purpose of capturing the performance and efficiency of the public sector for all new EU member states after the 1st of May 1994. The proposed indicator comprises data regarding education, public administration, health, economic stability, financial indexes, and income distribution.

The concept of public efficiency has been widely studied and has remained a subject of constant interest over the years, considering the dynamic of the crossed periods. Narbón-Perpiñá and De Witte's [24] bibliographical study shows that Afonso's model has been widely recognised in the literature regarding public efficiency, with input and output indicators being adapted according to the particularities of the studies, adding to the primary areas defined by Afonso et al. [23] outputs concerning communal services, recreational facilities, social services, public safety, markets, environmental protection, and business development.

Municipal efficiency has been a subject of analysis for the authors Balaguer-Coll et al. [25], who employed a two-step methodology, first to measure Spanish municipal efficiency, then to link this efficiency to economic growth, highlighting the importance of public efficiency for

sustainable development. Further applying the bases of Afonso et al. [23], Trabelsi and Boujel-bene [26] enrich the literature by also explaining the link between public spending efficiency and economic growth using the DEA methodology for measuring public efficiency, and a GMM methodology to further study the impact of efficiency on economic growth, in the case of 75 developing countries. The results were in line with the work of Balaguer-Coll et al. [25].

In relation to sustainable development goals, authors Ríos et al. [27] highlight that meeting sustainability goals may impose costs in terms of public efficiencies, especially in more nuanced areas such as social protection and equality. Public efficiency and its relation to accounting reforms were explored by Cuadrado-Ballesteros et al. [28], following the classic representation of Afonso et al. [23] in terms of public efficiency scores. The results highlight that accrual accounting is associated with increased efficiency in 22 European countries.

Analysing the concept of public efficiency from the perspective of digitalization and e-Governance, the study of Krejnus et al. [29] employs DEA methodology to determine the digital efficiency of the EU countries, stressing the need for further investigating the area of e-government, considering it a tool for achieving high levels of development and reducing the administrative burden.

Similar studies in the field, such as the paper of Cifuentes-Faura et al. [30], examine the importance of institutional transparency in the case of municipalities' performance. Applying several nonparametric methodologies but also stating their limitations, such as the sensibility of the results in relation to the variable selection, the authors argue that a more transparent environment in terms of economic activities and public policies leads to a more efficient administration.

Cárcaba et al. [31] highlight the positive relation between participation, accountability, and quality of life scores when analysing Spanish municipalities, however, the nexus between transparency and quality of life is still open to investigation. Following the same line of research, Gonzáles et al. [32] use quality of public governance as a dimension for constructing a quality-of-life index.

Furthermore, Poniatowicz et al. [33] conducted an econometric analysis to investigate the relationship between institutional characteristics related to the quality of governance and to the level of economic growth per unit of population in EU countries (including the UK). Their results show a positive link between indicators that measure the quality of governance (excluding political stability) and the degree of economic progress. The extent to which government quality contributes to increasing the quality of life was analysed by Helliwell et al. [18] on a sample of 157 states, and the results obtained confirm that improvements in the governance quality within policies over a reasonable timeframe may have a substantial impact on populations' general well-being and quality of life. Starting from the relationship between the forms of governance and efficiency in societies with and without a state, Angelini et al. [34] analysed how different forms of governance affect the level of well-being of each society.

All of the above, together with the study of Debnath and Shankar [35], laid the basis of our research question. The authors studied whether governance characteristics can increase population happiness, using quality of governance indicators as input variables. Grigoli and Ley [36] highlight that increased levels of governance are correlated with efficiency scores. The same authors point to the work of Rajkumar and Swaroop [15], who proved that in an economic environment characterized by low levels of governance quality, increased expenditures do not materialize in outputs. The government efficiency index has also been considered an input, a resource, in the study of Luna et al. [37], when studying the efficiency of e-government portals.

To summarize the review presented above, the topic of public efficiency has been explored in the current literature from many perspectives and views; however, to our knowledge, it has

rarely been investigated in connection to government quality, as an input factor. Thus, as it can be observed, and based on what we could find, there are no studies referring to and reporting on the nexus between governance, as a resource, and public sector performance, and then further investigating the factors that may impact this efficiency.

## Data and methodology

This research paper has two major objectives. The first purpose of this analysis is to highlight the significance of governance quality in terms of public sector efficiency, and the second is to identify the elements that influence and stimulate public efficiency. To achieve these goals, the methodology is divided into two main parts, the same as the methodological framework used by Moutinho et al. [2] in their study: performing a Data Envelopment Analysis (DEA) to determine the public efficiency for the EU27 countries, followed by applying a quantile regression to explain the efficiency scores obtained, using independent variables that portray population characteristics, as well as corruption, economic freedom, and government digitalization.

### Data envelopment analysis

DEA is a nonparametric technique that allows the calculation of efficiency scores by comparing a finite number of decision-making units (DMU), based on both the input and output data of each DMU. Considering that the public system cannot be deemed to be operating at an optimal scale [35], in this research paper, the BCC VRS model is applied.

The input-oriented approach of the DEA involves minimizing input variables by constraining the weighted amount of outputs, while the output-oriented approach assumes maximizing the results (the output) while constraining the level of inputs [38]. Because in this study the inputs reflect the quality of governance and the objective is to maximise the output represented by public efficiency, and considering that diminishing the quality of government is not something beneficial and desirable, the output-oriented BCC VRS model is applied to determine the efficiency score of each DMU, represented by an EU member state.

The analysis of the government quality's efficiency was conducted for all the 27 member states of the European Union, in the 2005–2020 period. This first part of the methodology makes use of Debnath and Shankar's [35] input methodology as well as the well-established model of Afonso et al. [23] when considering the output variables (Table 1).

In their paper, Debnath and Shankar [35] analyse the relationship between high government quality and elevated levels of happiness among the population, using quality of governance indicators as input variables, governmental efficiency being also an index considered a public resource in the study of Luna et al. [37]. Similar to the previously mentioned paper, governance as an input is measured using the WGI indicators. The governance quality variables provided by the World Bank are frequently used in the literature to capture governance at the state level, in relation to economic growth, public debt, wellbeing, or green growth [39–47]. Although the methodology of determining these particular indicators is being debated in the literature [48], WGI indicators remain regarded in the field of political and social science as a proxy that is effective in quantifying such an abstract and difficult to quantify concept, as governance [49].

The output variables follow the established model of Afonso et al. [23], widely employed in determining public efficiency. Several studies [50–59] reinforce this model by using the aggregate output index in the form of Public Sector Performance (PSP), which incorporates aspects of education, health, distribution, stability, infrastructure, and economic performance, with slight adaptations. Considering the remarks presented above, the choice of the output variables aims to cover all the key areas of the public sector in order to establish as complete a picture as possible. Furthermore, the vast majority of studies concerning public efficiency are developed

**Table 1. The variables used in the first part of the methodology.**

| Input variables | | | |
|---|---|---|---|
| **Variable name** | **Unit** | | **Source** |
| **Technical quality of governance** | Score (-2.5;2.5) | | World Bank |
| **Democratic quality of governance** | Score (-2.5;2.5) | | World Bank |
| **Governmental expenditures** | % GDP | | Eurostat |
| Output variables | | | |
| **Variable name** | **Unit** | **Public sector** | **Source** |
| **Tertiary education attainment** | % of total population between 25–64 | Education | Eurostat |
| **Infant survival rate** | Number of living new-borns/1,000 births | Health | Eurostat |
| **Life expectancy at birth** | Years | | Eurostat |
| **Income equality Inverse of Gini** | Points | Distribution | Eurostat |
| **Median Income** | EUR | | Eurostat |
| **Inverse of inflation** | % | Stability | World Bank |
| **GDP/capita** | Current prices, euro per capita | Economic performance | Eurostat |
| **Employment rate** | Persons in the labour force % of total population between 20–64 | | Eurostat |
| **Motorway's density** | Km/ 1000 km$^2$ | Infrastructure | Eurostat |
| **Share of renewable energy** | % Total energy consumption | Green infrastructure | Eurostat |

Source: authors' processing

based on the well-known input–output model, as highlighted by the work of Narbón-Perpiñá and De Witte [24].

For the present model, Afonso's model has been adapted, considering the particularities of our study. The administrative dimension considered by Afonso et al. [23] in their papers is replaced with both grey and green infrastructure as an essential element in public sector performance, in light of the fact that the administrative dimension is already regarded as an input element in this study.

Grey infrastructure is measured using the motorway density variable, road density being a relevant proxy for infrastructure according to the literature review of Narbón-Perpiñá and De Witte [24], in which the authors revealed that road infrastructure is one of the most used variables for infrastructure alongside street lightning [53, 60–62], while green infrastructure is measured by environmental performance captured by the share of renewable energy in total energy consumption, taking into account the current environmental and economic context, with accelerating climate change and Member States' efforts to combat this phenomenon.

For specific variables, their inversed versions were used to conduct this study (inverse of inflation, inverse of Gini), considering the DEA methodology, in which all the output indicators must fit the mathematical condition that higher values represent more favourable results, as stated by Chung [63].

By aggregating the standardised values of the variables described above, we were able to generate a single input variable and a single output variable. Each input variable, as well as each output variable, has an equal share in their index in this research. Thus, the analysis employs a BCC VRS output-oriented methodology, with 1 standardised input index and 1 standardised output index.

## Quantile regression

The development of our research is based on the relevant literature in the field when considering the determinants of public efficiency. Once public efficiency has been determined and quantified, the study's main focus is centred around analysing the determinants of efficiency,

**Table 2. The variables used in the regression part of the methodology.**

| Dependent variable | | | | |
|---|---|---|---|---|
| **Variable name** | **Unit** | **Abbreviation** | | **Source** |
| **Efficiency score** | Score (0;1) | *eff_score* | | Own calculation |
| Independent variables | | | | |
| **Variable name** | **Unit** | **Abbreviation** | **Expected influence** | **Source** |
| **Human development index** | Score (0;1) | *hdi* | + | United Nations Development Program |
| **Population density** | Persons per km2 | *pop_density* | +/- | Eurostat |
| **Old dependency ratio** | % of working age population | *old_depr* | - | Eurostat |
| **Net migration rate** | % | *migr* | +/- | Eurostat |
| **Corruption perception index (rescaled)** | Score (0;10) | *cpi_rescaled* | - | Transparency international |
| **Democracy index** | Score (0;10) | *demo_index* | + | V-Dem Institute-Department of Political Science, University of Gothenburg |
| **Economic freedom** | Score (0;10) | *ec_freed* | + | Fraser Institute—Economic freedom of the world (EFW) |
| **Trade openness** | % GDP | *trade* | + | World Bank |
| **Foreign direct investments** | % GDP | *fdi* | + | World Bank |
| **E-government** | % of all individuals that use the internet to interact with public authorities in the last 12 months | *egov* | + | Eurostat |

Source: authors' processing

with an emphasis on population characteristics, corruption, economic freedom, and governmental digitalization (Table 2).

When it comes to emphasising the determinants of public efficiency, in the second part of the paper dedicated to the study of public efficiency, Narbón-Perpiñá and De Witte [64] present a summary of the most used and relevant variables employed in the study of public efficiency, grouping them into six main categories of environmental variables, such as demographic, economic, political, financial, geographical, and institutional determinants, with the social determinants being the most used in the literature as highlighted by the authors. While examining the impact of each determinant on public efficiency, the authors note that the effects of these environmental variables differ from study to study, highlighting the importance of future studies in the area. This observation serves as the foundation of our selection process for the variables used in this part of the methodology.

According to the study of Hauner and Kyobe [59], population and demographic characteristics such as population density and population age are key drivers that contribute to government efficiency. Similar studies arguing the importance of population density on public efficiency or public spending are those of Alonso et al. [65], Rusmin et al. [66], Da Cruz and Marques [61], Afonso et al. [23], MacDonald [67], Holcombe and Williams [68], and Farnham [69]. Furthermore, numerous papers [59, 61, 65, 69, 70] tracked the relationship between population age and public performance. The impact of population density on efficiency can be discussed from different perspectives. In their article, Halkos and Tzeremes [71], pointed out Portnov and Errel's [72] remark on population density and its associated inequality, affecting governmental policies and socio-economic development. Moreover, Andrews and Entwistle [73], Lampe et al. [74], Da Cruz and Marques [61], Geys et al. [70], and Kalb et al. [75] point to the fact that a denser population increases the cost of public services due to the crowding effect, leading to inefficiency.

On the other hand, Alatawi et al. [76] found that public hospitals' efficiency scores are positively influenced by population density, considering that an increased population density comes with a higher demand for public services, which might result in increased health service output and higher efficiency scores, in line with the results of Radulovic and Dragutinovi´c [62]. Furthermore, a denser population can bring cost advantages to an economy, considering the concept of scale economies. However, this argument can be countered by the shortage of indivisible public goods with regard to a numerous population.

When it comes to population age and public efficiency, Yoshino and Miyamoto [77], throughout a DSGE model, and Radulovic and Dragutinovi´c [62], using a stochastic frontier approach, proved that population ageing has a powerful negative impact upon economies, affecting the efficiency of public policies. On the other hand, a younger population demands higher public services, implying higher public spending, which can negatively impact efficiency scores [64].

Migration or ethnic diversity is also an important variable considered by the literature in the discussion regarding public efficiency and economic development [74, 78]. The paper of Narbón-Perpiñá and De Witte [64] also points out the discussion regarding the migration rate in relation to public efficiency. The authors highlight a debate in the literature, emphasising the remarks of authors that demonstrate a positive correlation between migration and efficiency [74], due to the fact that even though public services increase in the short term, the effect on public spending will not be perceived immediately, and authors who proved that a more ethnically fragmented society is likely to be more inefficient [73].

Human development index (HDI) is widely used in economic research as a proxy for social development, being studied in relation to public efficiency and economic wealth, taking into account population education levels, standard of living, and life expectancy [79–84]. Looking from the education point of view, education, as part of the HDI, reflects an evolved, free, democratic society, with citizens aware of the power of their vote, of their rights and obligations. We can argue that through education, HDI contributes to the increase of institutional quality, which promotes efficient public financial policies to eliminate the waste of public resources and corruption. Moreover, a higher proportion of educated residents may lead to a competent labour force, contributing to public efficiency [61, 62, 64].

Furthermore, according to Narbón-Perpiñá and De Witte [64], Asatryan and De Witte [85], and Afonso et al. [23], citizens with higher incomes, by paying greater taxes, can hold the authorities accountable for the efficiency of services provided, leading to increased efficiency. On the other side, higher public resources and revenues could be associated with low interest from authorities in improving spending efficiency.

Corruption together with rent-seeking activities are also elements of particular interest when it comes to evaluating their impact on economic wealth and public spending efficiency, as highlighted by Dincă et al. [41], Fonchamnyo and Sama [86], Yogo [87], Hauner and Kyobe [59], and Afonso et al. [23]. The authors argue that in most cases, a high level of corruption deters economic growth and is correlated with low levels of public efficiency or low quality of public administration. Hauner and Kyobe [59] also studied the relationship between freedom captured by democracy, trade liberalisation and openness, and public sector performance. Fonchamnyo and Sama [86] utilised trade as an openness proxy, while Afonso et al. [23] portray freedom with the help of variables such as trust in politicians, trade openness, and transparency of institutions. Trade and foreign direct investments are also variables that Liberati [88] proposed in a very in-depth literature review regarding openness and FDI and their impact on the economy, welfare, public spending, and government size, presenting both negative and positive influences, centred mainly around the taxation of capital openness. The main conclusion pointed out in the above-mentioned paper is that both trade and FDI are negatively

associated with government size and also with government expenditures, which can manifest a positive impact upon public efficiency.

Digitalization is also a key area to take into consideration when it comes to public efficiency in the context of the accelerated growth of the IT industry and technologies from the last decade. Digitalization is believed to increase public efficiency, by saving time and work for both the population as well as for frontline bureaucrats, the latter having now more time to perform other tasks. However, as argued by Loberg [89], through what the author calls "cushioning" of the resources, bureaucrats may find it hard to transfer saved time into other specific tasks. Digitalization can be considered an opportunity for efficient governments to reduce the number of non-performing bureaucrats, with poorly defined tasks, and in positions that consume significant resources in terms of public funds. Considering all the above, digitalization is expected to manifest a positive influence on public efficiency scores in terms of governance quality.

The estimation techniques used are in line with those approached by Singh and Jha [90] and Moutinho et al. [2], the analysis centring around quantile regression estimation techniques to the detriment of standard linear regression (OLS) and panel data regressions. In their studies, the authors mentioned above point out the limitations of these classic mean-based estimation techniques, such as "having under or overestimate effects in heterogenous distribution" [90] or "failing to address the DEA efficiency score reliance problem", but also providing only a partial, average relationship between the independent variables and the efficiency scores, concentrating only at the mean of the distribution [2]. The quantile regression technique is considered to better interpret the relations between the endogenous and exogenous variables, considering the non-normal distributions, providing a better understanding in regard to these relations at different quantiles, being first developed by Koenker and Basset in 1978 [91].

Taking all the above into consideration, the regression equation reads as follows:

$$eff\_score =$$
$$\propto + \beta 1 \ hdi \ it + \beta 2 \ pop\_density \ it + \beta 3 \ old\_depr \ it + \beta 4 \ migr \ it + \beta 5 \ cpi\_rescaled \ it$$
$$+ \beta 6 \ demo\_index \ it + \beta 7 \ ec\_freed \ it + \beta 8 \ trade \ it + \beta 9 \ fdi \ it + \beta 10 \ egov \ it + \mu i$$
$$+ \varepsilon it;$$

where *eff_score* is the endogenous variable, and the exogenous independent variables are the following *hdi*, *pop_density*, *old_depr*, *migr*, *cpi_rescaled*, *demo_index*, *ec_freedom*, *trade*, *fdi*, *egov*, μi which captures the constant effect and particularities of EU countries i = 1,2. . .27 and εit as the error term.

## Results

### Efficiency scores

As explained in the methodology section above, the results were determined by processing the inputs and outputs in the 1 input, 1 output BCC VRS output-oriented scenario of Data Envelopment Analysis. The efficiency scores take values in the range of the 0–1 interval, where the most efficient countries in terms of governmental performance are closer to the upper value of the interval. Fig 1 provides a visual representation of these scores.

Following the analysis, each year, three countries are determined to be 100% efficient in terms of the quality of governance. The country that is considered efficient throughout the entire 2005–2020 period is Luxembourg, with a score of 1 each year.

Considered 100% efficient are also countries such as Romania in 10 of the 16 years in the study, Bulgaria in 7 of the 16 years, Croatia, reaching the maximum levels of efficiency during

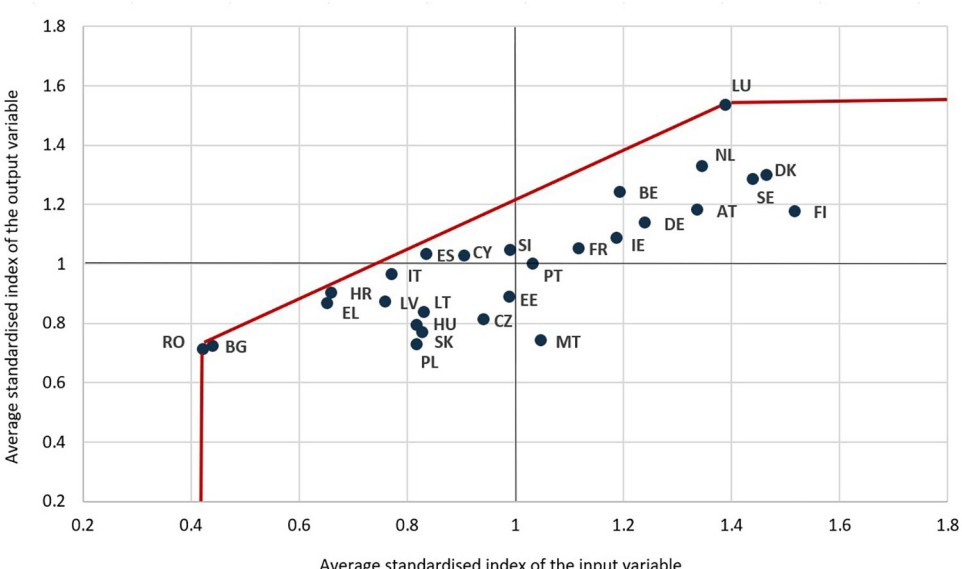

**Fig 1. Countries' distribution in relation to the efficiency frontier, according to the mean value of input and output indexes for the considered period.** Source: authors' processing.

the 2005–2008 period, Greece in 2014 and also in the 2016–2019 period, Spain in 2009 and 2010, and Cyprus at the end of 2020.

Alongside these states, high-efficiency scores are also found in countries such as Belgium, Cyprus, Italy, and Spain, occupying, on average, leading positions in the rankings regarding the quality of governance and its impact on public efficiency.

On the other side, the lowest efficiency scores are recorded during the whole period in the case of Malta, where the country failed to transform its efforts regarding the quality of governance into public sector performance. Governance efficiency problems highlighted by low-efficiency scores are also found in the cases of countries such as Czechia, almost throughout the whole period, Poland in the 2008–2016 period, and Estonia, starting from 2016.

Even if most states are relatively constant over the considered period in terms of efficiency scores, some states show significant changes, such as Croatia, Romania, Spain, and Greece, which show decreases in terms of the effectiveness of government quality over the performance of the public sector as a whole, while countries such as Cyprus, Bulgaria, Ireland, Portugal, Slovakia, Hungary, and Malta have improved their performance in terms of mathematical efficiency (Fig 2).

## Regression results

Prior to presenting the regression results, the key descriptive statistics of the considered variables as well as the correlative nature of the relationship between them were determined.

The correlation between the *eff_score* dependent variable and the independent variables is weak, without exceeding the value of 0.32 in absolute terms, and positive in relation to the *hdi*, *old depr*, and *cpi_rescaled* variables.

The performed variance inflation factor (VIF) test resulted in a mean VIF of 2.59, thereby demonstrating the validity of the regression model developed and also the validity of the considered variables, proving that the proposed model has no multicollinearity [92].

The classic OLS and panel data regressions such as the random effects model (REM), fixed effects model (FEM) and fixed effects model with dummy variables (LSDV) were performed.

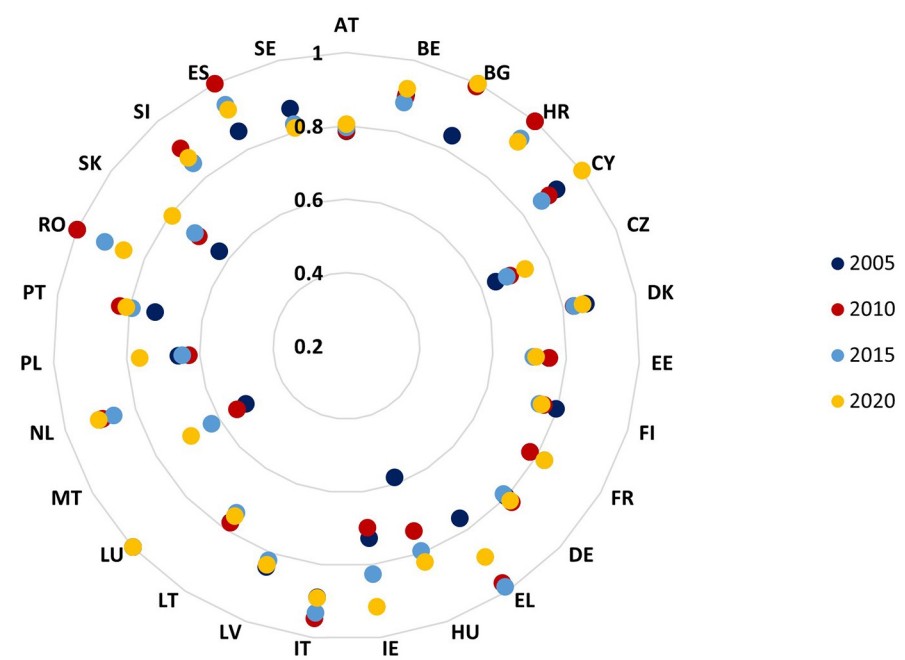

**Fig 2. Governance efficiency scores for the EU27 sample in the years 2005, 2010, 2015, and 2020.** Source: authors' processing.

The results of the statistical tests show that, if we use the regression methodology specific to panel databases, the FEM together with a panel-corrected standard errors regression (PCSE) would be the most appropriate methods of analysis for our data.

However, analysing the histogram of the efficiency scores and the quantile distribution, we can conclude that they do not follow a balanced distribution, posing a considerable obstacle to the use of mean-based regression techniques. Therefore, quantile regression is a proper technique to assess how the quality of human resource or population characteristics, corruption, freedom and democracy, and government digitalization affect the EU countries in accordance with the distribution of their efficiency scores.

When performing the quantile regression, both the classical pooled quantile regression and the heteroskedasticity robust standard errors quantile regression were performed, in line with the methodology presented in the studies of Moutinho et al. [2] and Singh and Jha [90].

One of the main conclusions that can be drawn from applying the classical approach of quantile regression (Table 3) is that except for the first quantile, the human development index is significantly and positively correlated with government efficiency scores. Moreover, the positive impact of the *hdi* on the *eff_score* is much more pronounced in the upper quantiles, in countries with high levels of efficiency scores, compared to the 10[th] or 25[th] quantiles.

The EU member states' public efficiency scores were negatively impacted by population density as an indicator of human resources, the negative impact being relatively constant across the studied quantiles. The old dependency ratio has a significant positive impact on efficiency in the first three quantiles, following that in the case of countries with higher efficiency scores not to exert a significant influence. Migration, however, manifests a positive and significant influence in terms of efficiency for all quantiles.

**Table 3. Pooled quantile regression results.**

| Dependent variable: Efficiency score | | | | | |
|---|---|---|---|---|---|
| Independent variables | (1) | (2) | (3) | (4) | (5) |
| | Q10 | Q25 | Q50 | Q75 | Q90 |
| hdi | 0.5131192 | 0.7406806*** | 1.1531351** | 1.1425130*** | 0.9646113*** |
| | (0.4312871) | (0.2854137) | (0.4718966) | (0.3145746) | (0.2725614) |
| pop_density | -0.0001847*** | -0.0001598*** | -0.0002303*** | -0.0001389*** | -0.0000676** |
| | (0.0000435) | (0.0000288) | (0.0000476) | (0.0000317) | (0.0000275) |
| old_depr | 0.0111568*** | 0.0075765*** | 0.0074199** | 0.0010947 | -0.0017034 |
| | (0.0026423) | (0.0017486) | (0.0028911) | (0.0019273) | (0.0016699) |
| migr | 0.0025407* | 0.0016335* | 0.0027860* | 0.0025896** | 0.0019919** |
| | (0.0014637) | (0.0009686) | (0.0016015) | (0.0010676) | (0.0009250) |
| cpi_rescaled | 0.0067721 | 0.0004842 | 0.0104314 | 0.0147451* | 0.0140144* |
| | (0.0116024) | (0.0076782) | (0.0126949) | (0.0084626) | (0.0073324) |
| demo_index | -0.0933311 | -0.1377786 | -0.3006210** | -0.4262651*** | -0.4192185*** |
| | (0.1325494) | (0.0877175) | (0.1450301) | (0.0966796) | (0.0837675) |
| ec_freed | -0.0655574* | -0.0875125*** | -0.1007380*** | -0.0932569*** | -0.0901969*** |
| | (0.0341084) | (0.0225720) | (0.0373200) | (0.0248782) | (0.0215556) |
| trade | 0.0002683 | -0.0000972 | 0.0000470 | 0.0004033*** | 0.0002897** |
| | (0.0002004) | (0.0001326) | (0.0002193) | (0.0001462) | (0.0001267) |
| fdi | 0.0005286* | 0.0004920*** | 0.0004403 | -0.0001455 | -0.0001669 |
| | (0.0002818) | (0.0001865) | (0.0003083) | (0.0002055) | (0.0001781) |
| egov | -0.0003945 | -0.0013127** | -0.0019909** | -0.0006101 | -0.0004751 |
| | (0.0008037) | (0.0005319) | (0.0008794) | (0.0005862) | (0.0005079) |
| Constant | 0.5200242 | 0.7874399*** | 0.6871970 | 0.8572628** | 1.0936586*** |
| | (0.4571983) | (0.3025610) | (0.5002475) | (0.3334738) | (0.2889365) |
| Observations | 323 | 323 | 323 | 323 | 323 |
| Pseudo R2 | 0.2833 | 0.2273 | 0.1496 | 0.1922 | 0.1990 |

Standard errors in parentheses

*** p<0.01

** p<0.05

* p<0.1

Source: authors' processing

Counterintuitive results are met in the cases of perceived corruption, democracy, economic freedom, and digitalization. Perceived corruption (*cpi_rescaled*) manifests a positive impact on public efficiency, however the significance is captured only in the 75th and 90th quantiles.

Democracy and economic freedom have a similar impact on EU countries' efficiency, exerting negative and significant influences along all considered quantiles, except for the first two quantiles in the case of democracy index variable, where the influence is not statistically significant. The negative impact of *demo_index* variable is much more accentuated in the countries from superior quantiles, while *ec_freed*'s impact is felt more strongly in the median and upper quantiles, compared to the other considered levels. Digitalization, through the *egov* variable, exerts a negative impact on all quantiles of efficiency scores, while a significant impact is revealed for the 25th and 50th quantiles.

The positive and significant influence of trade openness on governance efficiency is emphasised in the upper quantiles of efficiency scores, while the positive and significant effects of foreign direct investment are captured in the European Union countries with the lowest public efficiency scores.

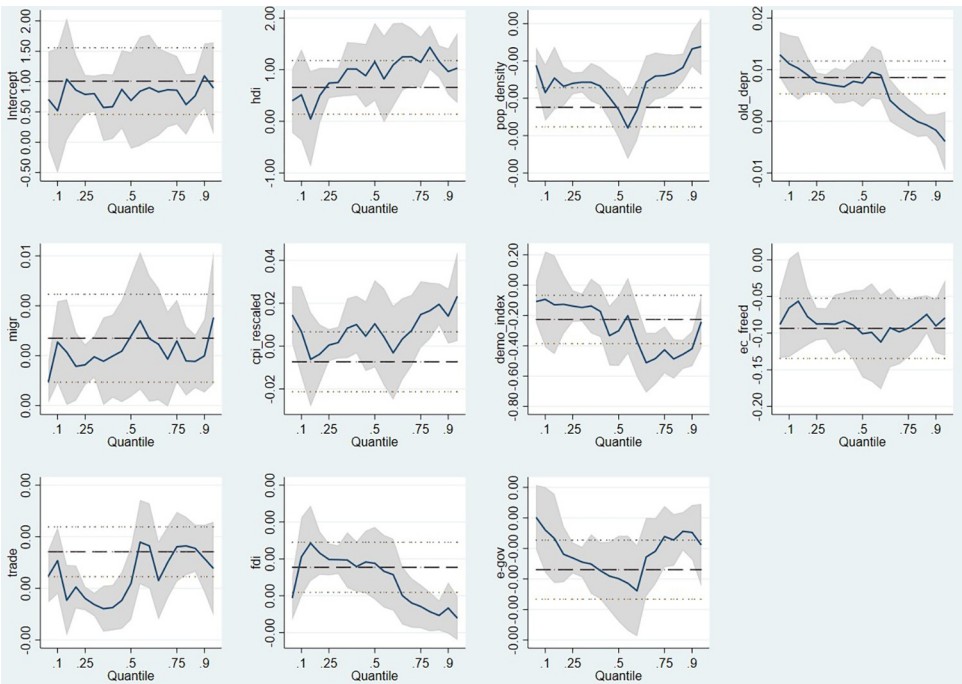

**Fig 3. Results of the quantile and OLS regressions—the coefficients estimated on different quantiles.** Source: authors' processing.

Furthermore, when controlling for heteroscedasticity and applying the heteroskedasticity robust standard errors quantile regression [93], the main results of the regression are maintained, with the only differences in the heteroskedasticity robust standard errors quantile regression being the slight change in significance levels of the independent variables for specific quantiles, except for the old dependency ratio, corruption, and trade variables.

The quantile regression results may also be visualised with the help of Fig 3, where the effects of each independent variable and their variations at different quantiles are illustrated, in comparison with the OLS estimated coefficients represented by the horizontal line [2].

## Discussions

The results presented above show the differences across the EU countries in terms of governance efficiency.

Luxembourg, the country that is considered efficient throughout the entire period, is the EU state with the highest median income, analysed in comparison to the EU average, in the top three countries regarding tertiary education attainment, alongside Finland and Ireland, governance quality, economic performance per capita, and having the most developed infrastructure in terms of kilometres of motorways per square km, alongside the Netherlands and Belgium, at the end of 2020. However, in the case of Romania, Bulgaria, Greece, and Croatia, an important note must be made regarding the methodology. The reduced levels of resources in terms of governance quality are being seen as efficiency benefits, considering that similar, but still lower output levels as in other EU countries, are obtained. Thus, we can admit that these countries can be considered efficient only from a mathematical and empirical point of view, since lower levels of governance quality cannot be considered advantageous in the real economy.

On average, the efficiency of the EU countries in terms of governance quality and public sector performance shows an upward trend in the 2005–2011 period, from an average score of 0.8025 points to 0.8360 points, followed by a slight decrease to 0.8131 points in 2014. The maximum level of efficiency was achieved at the end of 2019, when the governments of the EU states managed to make the most efficient use of their management tools to contribute to the performance of the public sector. However, at the end of 2020, the efficiency diminished, a decrease attributable to the start of the Covid-19 pandemic, which also affected the EU member states. Thus, we can say that at the end of 2020 (when an efficiency score of 0.8474 points was obtained), the effect on public sector performance at the European level could have increased by 15.26%, given the current governance. Furthermore, on average, the veteran EU countries that joined the European Union structure before 2004 record higher efficiency scores than the EU average throughout the analysis period.

Analysing the regression results, we can conclude that there are a few significant discrepancies in terms of the expected influence of the independent variables and their actual obtained impact upon public efficiency. Furthermore, it is important to note that the determinants of government efficiency differ depending on the efficiency quantiles.

The hypothesis that the human development index leads to a more effective performance of the public sector in terms of governance is accepted with a probability greater than 95%, for all quantiles, except for the first one. Thus, our results are in line with those of Niswaty et al. [83] and Rusmin et al. [66], who proved that an increase in HDI is correlated with an increased level of government performance. Moreover, relevant studies show that an increased level of HDI is associated with reduced tax evasion [82], economic growth [81], and reduced poverty levels, as presented in the case study of the South Sulawesi region of Indonesia, conducted by Fahrika et al. [80] and Dahliah and Nur [79], which beneficially impacts public sector performance.

Even though it could have been argued that a higher population density is likely to increase public sector performance by lowering service delivery costs through the economies of scale concept, our results show the opposite. Population density can also be viewed as a stimulant, especially in the case of public service provision, as argued by Delgado-Antequera et al. [94] in the analysis of eco-efficiency in terms of municipality waste, and Alatawi et al. [76] in terms of health services. Our results, however, suggest that this is not the case in terms of public sector efficiency in its entirety. The number of inhabitants per square kilometre manifests a negative and significant impact upon all quantiles of efficiency, in accordance with Halkos and Tzeremes's [71] conclusion, which proved a negative influence of population density on the Greek public sector's efficiency, and also with the statement of Moreno-Enguix and Lorente Bayona [95], that a higher level of population density is associated with lower levels of public efficiency. Moreover, our study adds to the literature in the field, endorsing the work of authors such as Andrews and Entwistle [73], Lampe et al. [74], Da Cruz and Marques [61], Geys et al. [70], and Kalb et al. [75], who point to the fact that a denser population is responsible for the crowding effect that manifests in the case of public services, leading to inefficiency.

Our results regarding the age dependency ratio prove that for lower quantiles of efficiency scores, a more mature population significantly contributes to an increased level of government efficiency. These results are in line with the remarks of Hauner and Kyobe [59], who argued that a younger population decreases the efficiency of the public sector, considering its rising costs, while an older population increases it. The results contradict the conclusions of the authors Yoshino and Miyamoto [77] and Radulovic and Dragutinoviˊc [62], proving that population ageing does not have a significant negative impact on public policies' effectiveness. Narbón-Perpiñá and De Witte [64] argue that with age, older people are more involved and concerned about government performance and are more involved in local public authorities.

The net migration rate is one of the independent variables that manifests a positive significant impact for all quantiles in our study, thus being an important factor to take into consideration in improving public efficiency for all countries in the dataset. Considering the results obtained, our study confirms the conclusions of Lampe et al. [74], who argue that the implications of the costs generated by new immigrants are not significantly felt by the public administration, on the contrary, they contribute to economic development and economic efficiency, at least in the short term.

In regard to corruption, the obtained results are counterintuitive. In our study, corruption manifests a significant positive impact on public efficiency starting from the 75th quantile. When analysing the literature in this specific domain, Hauner and Kyobe [59] point out that higher levels of efficiency are linked to lower levels of corruption. On the same note regarding the detrimental effects of corruption, Gamberoni et al. [96] demonstrated a positive relationship between corruption growth and the misallocation of productivity factors, which is more prevalent in less developed and not so well-regulated countries. Furthermore, Sarkar and Hasan [97] concluded that low levels of corruption imply an increase in both the volume and productivity of investments, which leads to economic growth. Conducting an analysis based on regression techniques for 166 countries over the period from 2004 to 2017, Malyniak et al. [98] proved that corruption impacts public spending efficiency. However, on the other hand, Thach et al. [99] point out that corruption can also be seen as a positive factor in economic growth and institutional and private efficiency, but only if it is referred to as avoiding bureaucratic processes. This argument is also sustained by Ertimi and Saeh [100] when analysing the link between corruption and economic development in the literature, arguing that the costs of eliminating corruption increase as the level of perceived corruption decreases, pointing to a threshold that, if exceeded, can make eliminating corruption more costly for society than the effects of corruption itself. Another solid observation is made by Drury et al. [101], who state that economic progress is not affected by corruption in nations with a mature democracy system, which in our case may be valid for the EU Member States considering their high levels of democracy in contrast with other world states, whereas non-democratic countries have significant economic disadvantages as a result of this phenomenon.

As pointed out in the results section of the paper, the democracy and economic freedom variables also manifest a counterintuitive influence on public efficiency scores. The obtained results regarding the democracy and economic freedom variables are in contrast with the findings of Moreno-Enguix and Lorente Bayona [95], who proved a positive correlation between democracy levels and high levels of efficiency in terms of public expenditures, which was also pointed out in our study by the correlation matrix, strong democracies having low levels of corruption, mainly due to their educated population and their desire for a more transparent governance. Asatryan and De Witte [85] proved, using the example of Bavaria, that more inclusive, open, and citizen-oriented governance through direct and transparent "decision-making mechanisms" may lead to more efficient governance. Moreover, Roessler and Schmitt [102] studied the efficiency of the health public system, proving that democratic governments strive more than non-democratic governments to eliminate inefficiencies through their policies. However, the arguments of Roessler [103] sustain our results. The author argues that democracy can harm public systems' efficiency, explaining that in less developed countries, the kleptocratic behaviour of governments is stronger the higher the level of public investment so that they remain in power. In other words, using a panel data regression analysis, the author proved that democracy enhances the provision of goods in relatively wealthy nations, whereas it decreases the provision of goods in underdeveloped nations.

As expected, trade openness positively impacts public efficiency, as expected, in line with the results obtained by Afonso et al. [23] and Sikayena et al. [104], who proved that trade

openness is positively associated with efficiency for both the education as well as for the health sector. However, trade openness loses its significance when analysing its impact on lower quantiles. The negative impact of trade openness is only captured in the 25[th] quantile, however, not on a significant level, in line with the results of Fonchamnyo and Sama [86] and Hauner and Kyobe [59]. The fact that FDI correlates positively and significantly with public efficiency for the first two quantiles is consistent with the results of Zhang et al. [105], who demonstrated that FDI is positively correlated with government efficiency in terms of environmental spending. Furthermore, the study by Wu and Lin [106] also concludes that trade openness and foreign investments can hinder government expansion and excessive intervention, leading to more efficiency-related public policies.

In the present work, digitalization has a negative influence on public efficiency for all levels of government efficiency scores, similar to the study of Doran et al. [107], who proved that the implemented online public services negatively influenced public efficiency mainly due to the lack of usage. To better understand how digitalization impacts public administration and public services, Andersson et al. [108] proved that through digitalization, public service is "fundamentally changed" and argue that while digitalization has the potential to minimise the complexity of public administration-related operations, at the same time it can decrease service quality, arguing that human interaction between citizens and professionals is what generates real service value. Digitalization offers the opportunity for efficient governments to reduce the number of non-performing bureaucrats, with poorly defined tasks, and in positions consuming large amounts of public funds, however, an important point made by Maiti et al. [109] must be taken into account in the analysis of digitalization's impact on public services. The authors point out that the impact of technology also depends on other institutional assets and conditions, and policies should not only promote digitalization as such, but also ensure its complementarity with country-specific elements. For example, the paper of Doran et al. [107] proved that for public efficiency to increase, more important than the digitalization of the systems themselves is the ability of the population to use these systems.

## Conclusions

The present paper has been developed starting from two questions: How significant is the quality of governance in assessing the efficiency of the public sector as a whole? What are the drivers of efficiency in the public sector, with an emphasis on human resources and economic freedom, democracy, corruption, and digitalization?

The main takeaway points of this study are that the quality of governance and its effectiveness, as illustrated by the scores determined by the DEA analysis, reveal that, on average, the EU member states are relatively ineffective, with only a few countries recording elevated levels of effective governance from a mathematical point of view, whose policies have resulted in increasing public efficiency. Thus, there is room for improvement and reform of current public policies even if public efficiency in the EU has increased in the studied period.

When analysing the determinants that can affect the performance of the public sector, our study, using the quantile regression technique, pointed out that the impact and significance of these determinants vary depending on the efficiency score quantile.

The net migration rate variable manifests a significant positive impact on efficiency scores for all levels of public efficiency, whilst variables such as population density, and economic freedom influence public efficiency significantly in a negative manner for all quantiles, as presented. Positive impacts are also emphasised by the human development index and perceived corruption for all quantiles, while, on the other hand, democracy index and digitalization negatively influenced public efficiency scores.

From a significance point of view, it is important to note that all variables show changes in statistical significance depending on the studied quantile, emphasising the importance of determinants of public performance in relation to the level of efficiency in EU states. For example, in the countries with the lowest efficiency scores from our sample, population density together with economic freedom impacts negatively public efficiency scores at a level of 1% significance threshold and 10% respectively, while the variables that contribute to an increased public efficiency are old dependency ratio (p-value $<0.01$), migration rate (p-value $< 0.1$) and foreign direct investments (p-value$<0.1$). For the 90[th] quantile, the determinants of public efficiency change. In comparison with the 10[th] quantile, the foreign direct investment variable and the old dependency ratio lost their statistical significance, while human development index, trade, corruption, and democracy index are now statistically significant elements that impact public efficiency in the most efficient countries in terms of governance in the EU structure.

## Limitations

The main limitation of the current study refers to the restricted sample size in terms of the considered period for the analysis, although statistically significant. Given the multitude of variables considered in this analysis, both for the determination of efficiency scores and for the analysis of the driving factors, it was not possible to identify a wider and more current period for the completion of the database, so that the database was composed for the limited period of 2005–2020. Moreover, looking at the factors that have significant correlations between them can lead to the creation of numerous independent econometric models, which can all be studied individually to identify the isolated influence of the relevant variables on governance efficiency.

Moreover, for future research, an in-depth subnational and regional approach for examining public efficiency from a quality of governance approach should be considered, given the increasing regional discrepancies encountered in the regions of the European Union.

## Policy recommendations

The main results, namely that the quality of governance can be considered an important resource in the analysis of public performance, and that human resources or demographic characteristics, freedom, democracy, corruption, and digitalization have an impact on this efficiency, are important points to consider not only for policymakers but also for society, researchers, and the academic community. Governance is a key factor in ensuring the performance of the public sector, so reform policies should aim at improving both the technical and democratic dimensions of public institutions, so that public resources are allocated efficiently and transparently. As our results suggest, this can only be achieved with the support of a population that is actively involved in the development of the state and has a low tolerance for political parties that are only interested in achieving their own objectives. Furthermore, our study points out that policymakers should first and foremost be aware of local and national characteristics and take them into account when developing specific policy reforms, considering the differences in statistical significance of macroeconomic determinants in relation to the level of public efficiency in the EU.

## Supporting information

**S1 Fig. Histogram of the efficiency scores.** Source: authors' processing.
(DOCX)

**S2 Fig. Quantile distribution of the dependent variable eff_score.** Source: authors' processing.
(DOCX)

**S1 Table. Efficiency scores of the EU27 countries in terms of good governance and public efficiency, obtained using DEA methodology.** Data source: authors' processing.
(DOCX)

**S2 Table. EU27 states' rankings by efficiency scores obtained using DEA methodology.** Data source: authors' processing.
(DOCX)

**S3 Table. Descriptive statistics for the EU 27 sample.** Data source: authors' processing.
(DOCX)

**S4 Table. Correlation matrix.** Data source: authors' processing.
(DOCX)

**S5 Table. OLS and panel data regression results.** Standard errors in parentheses. *** p<0.01, ** p<0.05, * p<0.1. Source: authors' processing.
(DOCX)

**S6 Table. Heteroskedasticity robust standard errors quantile regression results.** Standard errors in parentheses. *** p<0.01, ** p<0.05, * p<0.1. Source: authors' processing.
(DOCX)

## Author Contributions

**Conceptualization:** Camelia Negri, Gheorghița Dincă.

**Data curation:** Camelia Negri.

**Formal analysis:** Camelia Negri, Gheorghița Dincă.

**Investigation:** Camelia Negri.

**Methodology:** Camelia Negri.

**Project administration:** Camelia Negri, Gheorghița Dincă.

**Resources:** Camelia Negri, Gheorghița Dincă.

**Software:** Camelia Negri.

**Supervision:** Gheorghița Dincă.

**Validation:** Gheorghița Dincă.

**Visualization:** Camelia Negri.

**Writing – original draft:** Camelia Negri, Gheorghița Dincă.

**Writing – review & editing:** Camelia Negri, Gheorghița Dincă.

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
