## [Decision Letter · Decision Letter 0]

26 Jul 2023

PONE-D-23-15310Public sector’s efficiency as a reflection of governance quality, an European Union studyPLOS ONE

Dear Dr. Dinca,

Thank you for submitting your manuscript to PLOS ONE. After careful consideration, we feel that it has merit but does not fully meet PLOS ONE’s publication criteria as it currently stands. Therefore, we invite you to submit a revised version of the manuscript that addresses the points raised during the review process.

We look forward to receiving your revised manuscript.

Kind regards,

Javier Cifuentes-Faura

Academic Editor

PLOS ONE

Journal Requirements:

**Additional Editor Comments:**

You should take into account all the comments from the reviewers.

I also recommend you update the recent literature on public sector efficiency and DEA. You can cite in your paper the following ones:

Cifuentes-Faura, J., Benito, B., Guillamón, M. D., & Faura-Martínez, Ú. (2023). Relationship between Transparency and Efficiency in Municipal Governments: Several Nonparametric Approaches. Public Performance & Management Review, 46(1), 193-224.

Trabelsi, N., & Boujelbene, Y. (2022). Public Sector Efficiency and Economic Growth in Developing Countries. Journal of the Knowledge Economy, 1-20.

Ríos, A. M., Guillamón, M. D., Cifuentes‐Faura, J., & Benito, B. (2022). Efficiency and sustainability in municipal social policies. Social Policy & Administration, 56(7), 1103-1118.

Cuadrado-Ballesteros, B., Bisogno, M., & Vaia, G. (2022). Public-Sector Accounting Reforms and Governmental Efficiency: A Two-Stage Approach. The International Journal of Accounting, 57(04), 2250017.

Reviewers' comments:

Reviewer's Responses to Questions

**Comments to the Author**

1. Is the manuscript technically sound, and do the data support the conclusions?

Reviewer #1: Yes

Reviewer #2: Partly

2. Has the statistical analysis been performed appropriately and rigorously? 

Reviewer #1: Yes

Reviewer #2: Yes

3. Have the authors made all data underlying the findings in their manuscript fully available?

Reviewer #1: Yes

Reviewer #2: Yes

4. Is the manuscript presented in an intelligible fashion and written in standard English?

Reviewer #1: Yes

Reviewer #2: Yes

5. Review Comments to the Author

Reviewer #1: Dear Authors,

Congratulations for your interesting research. I have some suggestions on how to make your text more attractive

The introduction needs redrafting. The main problem is the epistemological structure (why the article was conceived and how the study was developed). I suggest the following structure of objectives: (i) research gap; (ii) research question; (iii) purpose of the article; (iv) intermediate objectives ; (v) assumptions or hypo; and (vi) research method. This structure must appear in the introduction.

I propose not to describe what parts the article contains in turn. This is obvious.

The research gap (Literature review) must be created by a systematic literature review that provides 'holes' in the state of knowledge on the topic. I believe that a full review should not be done, but an analysis of about 5-8 studies on the topic under discussion. You can find some examples, which will show the relevance of the issue, as it is indeed a topic of current, relevant research. At the end of the justification you should write something like: According to what we were able to find, there are no studies referring and reporting on ... With this you have therefore proven that the issue is relevant, and you have also proven that your study does indeed fill a research gap.

Reviewer #2: The novel features of the research (methods, variables used etc) have to be better explained in the introduction. It is not clear the added value of this paper compared with previous research in the field.

The two-stages of the empirical analysis are meaningful and complement each other. The description of the methodological features of both methods is concise, but orients the reader especially if he is already familiarized with these methods.

However, I feel that the main drawback of this analysis lies in the variables' selection process. In both cases (input and output variables for DEA, respectively explanatory variables for the quartile regression) there is no convincing substantiation of the choice of these particular variables. Why these and not others ? What is their informational content, their theoretical/intuitive relationship with the dependent variables ?

For example, the choice of input and output variables included in the DEA model specification is highly influencing the computation of the efficiency scores. Any variable change may significantly alter the scores and hence the efficiency status of a country. This part of the study needs to be better explained, in order to support the reliability and soundness of the findings.

The figures in the Annexes have no title.

6. PLOS authors have the option to publish the peer review history of their article (what does this mean?). If published, this will include your full peer review and any attached files.

Reviewer #1: No

Reviewer #2: No

---

## [Author Response · Author response to Decision Letter 0]

19 Aug 2023

Response to Editor and Reviewers Comments 

Comments on the editor’s and reviewers’ reports for the paper 

Public sector’s efficiency as a reflection of governance quality, an European Union study

 Our comments refer to the suggestions and recommendations the referees made in order to improve the quality of the research paper, and not to the positive remarks they made. We have to mention that most requirements were fulfilled in this new version of the paper.

Dear Editor, Dear Reviewers, 

First of all, we would like to express our gratitude for the work you dedicated for investigating our research, identifying the points for improvement, and suggesting ways for achieving that. We are fully aware that your suggestions and recommendations are very important for improving our research and the way it is presented in this article.

Thank you, and we hope we have answered all your suggestions and recommendations and improved our research.

Journal Requirements

1. Please ensure that your manuscript meets PLOS ONE’s style requirements, including those for file naming. The PLOS ONE style, templates can be found at

Comment: The manuscript and all the other additional files have now been carefully formatted to meet PLOS ONE’s style requirements.

2. PLOS requires an ORCID iD for the corresponding author in Editorial Manager on papers submitted after December 6th, 2016. Please ensure that you have an ORCID iD and that it is validated in Editorial Manager. 

Comment: The correspondence author's ORCID iD has now been validated in Editorial Manager.

Comment: Captions for supporting information files have been added at the end of the manuscript, in the “Supporting Information” section (rows 1084-1098). 

Additional Editor Requirements

1. You should take into account all the comments from the reviewers.

Comment: As we will clarify below, all reviewers' recommendations were taken into account while revising the manuscript, and all requirements were fulfilled.

2. I also recommend you update the recent literature on public sector efficiency and DEA. You can cite in your paper the following ones:

• Cifuentes-Faura, J., Benito, B., Guillamón, M. D., & Faura-Martínez, Ú. (2023). Relationship between Transparency and Efficiency in Municipal Governments: Several Nonparametric Approaches. Public Performance & Management Review, 46(1), 193-224.

• Trabelsi, N., & Boujelbene, Y. (2022). Public Sector Efficiency and Economic Growth in Developing Countries. Journal of the Knowledge Economy, 1-20.

• Ríos, A. M., Guillamón, M. D., Cifuentes‐Faura, J., & Benito, B. (2022). Efficiency and sustainability in municipal social policies. Social Policy & Administration, 56(7), 1103-1118.

• Cuadrado-Ballesteros, B., Bisogno, M., & Vaia, G. (2022). Public-Sector Accounting Reforms and Governmental Efficiency: A Two-Stage Approach. The International Journal of Accounting, 57(04), 2250017.

Comment: We have improved the literature review section regarding public efficiency and DEA with the recommended studies, and we have also enriched the literature with other relevant papers in this field, focusing on more recent studies (rows 143-182 and rows 195-203), such as:

• Cifuentes-Faura J, Benito B, Guillamón MD, Faura-Martínez Ú. Relationship between Transparency and Efficiency in Municipal Governments: Several Nonparametric Approaches. Public Performance & Management Review. 2023 Jan 2;46(1):193-224.

• Ríos AM, Guillamón MD, Cifuentes‐Faura J, Benito B. Efficiency and sustainability in municipal social policies. Social Policy & Administration. 2022 Dec;56(7):1103-18.

• Trabelsi N, Boujelbene Y. Public Sector Efficiency and Economic Growth in Developing Countries. Journal of the Knowledge Economy. 2022 Dec 6:1-20.

• Cuadrado-Ballesteros B, Bisogno M, Vaia G. Public-Sector Accounting Reforms and Governmental Efficiency: A Two-Stage Approach. The International Journal of Accounting. 2022 Dec 19;57(04):2250017.

• Krejnus M, Stofkova J, Stofkova KR, Binasova V. The Use of the DEA Method for Measuring the Efficiency of Electronic Public Administration as Part of the Digitization of the Economy and Society. Applied Sciences. 2023 Mar 13;13(6):3672.

• Cárcaba A, González E, Ventura J, Arrondo R. How does good governance relate to quality of life? Sustainability. 2017;9: 631.

• González E, Cárcaba A, Ventura J. Weight constrained DEA measurement of the quality of life in Spanish municipalities in 2011. Social Indicators Research. 2018;136: 1157–1182.

• Grigoli F, Ley ME. Quality of government and living standards: adjusting for the efficiency of public Spending. International Monetary Fund; 2012.

• Luna DE, Gil-Garcia JR, Luna-Reyes LF, Sandoval-Almazán R, Duarte-Valle A. Using data envelopment analysis (DEA) to assess government web portals performance. Proceedings of the 13th annual international conference on digital government research. 2012. pp. 107–115.

Moreover, the literature review section has been restructured, following a more systematic approach, starting from the concept of governance and processing to the quality of governance, then followed by what efficiency means, which papers on public efficiency are similar, and what they deal with, and concluding with a paragraph dedicated to the research gap (rows 204-210). 

 

Independent Review Report, Reviewer 1

Dear Authors,

Congratulations for your interesting research. I have some suggestions on how to make your text more attractive

1. The introduction needs redrafting. The main problem is the epistemological structure (why the article was conceived and how the study was developed). I suggest the following structure of objectives:

(i) research gap; 

(ii) research question; 

(iii) purpose of the article; 

(iv) intermediate objectives; 

(v) assumptions or hypo; and

(vi) research method. 

This structure must appear in the introduction.

Comment: We thank you so much for this suggestion. The introduction section has been redrafted, following the proposed structure of objectives (rows 47-91), focusing more on highlighting in a clear manner the epistemological structure, the novelty of the study and the research aim.

2. I propose not to describe what parts the article contains in turn. This is obvious.

Comment: We have removed the aforementioned paragraph.

3. The research gap (Literature review) must be created by a systematic literature review that provides holes; in the state of knowledge on the topic. I believe that a full review should not be done, but an analysis of about 5-8 studies on the topic under discussion. You can find some examples, which will show the relevance of the issue, as it is indeed a topic of current, relevant research.

At the end of the justification you should write something like: According to what we were able to find, there are no studies referring and reporting on ... With this you have therefore proven that the issue is relevant, and you have also proven that your study does indeed fill a research gap.

Comment: We have improved the literature review section, focusing our analysis on more recent studies regarding public efficiency and DEA (rows 143-182 and rows 195-203), such as: 

• Cuadrado-Ballesteros B, Bisogno M, Vaia G. Public-Sector Accounting Reforms and Governmental Efficiency: A Two-Stage Approach. The International Journal of Accounting. 2022 Dec 19;57(04):2250017.

• Krejnus M, Stofkova J, Stofkova KR, Binasova V. The Use of the DEA Method for Measuring the Efficiency of Electronic Public Administration as Part of the Digitization of the Economy and Society. Applied Sciences. 2023 Mar 13;13(6):3672.

• Cárcaba A, González E, Ventura J, Arrondo R. How does good governance relate to quality of life? Sustainability. 2017;9: 631.

• González E, Cárcaba A, Ventura J. Weight constrained DEA measurement of the quality of life in Spanish municipalities in 2011. Social Indicators Research. 2018;136: 1157–1182.

• Grigoli F, Ley ME. Quality of government and living standards: adjusting for the efficiency of public Spending. International Monetary Fund; 2012.

• Luna DE, Gil-Garcia JR, Luna-Reyes LF, Sandoval-Almazán R, Duarte-Valle A. Using data envelopment analysis (DEA) to assess government web portals performance. Proceedings of the 13th annual international conference on digital government research. 2012. pp. 107–115.

Moreover, the literature review section has been restructured, to follow a more systematic and focused approach, focusing in the end on the research gap. 

The argumentation of employed variables has been moved to the data and methodology section. The literature opens with the concept of governance, followed by the quality of governance, what efficiency means, analysis of the similar papers on public efficiency and what they deal with and what their limitations are, and closing the section with a paragraph dedicated to the research gap (rows 204-210). 

 

Independent Review Report, Reviewer 2

1. The novel features of the research (methods, variables used etc.) have to be better explained in the introduction. It is not clear the added value of this paper compared with previous research in the field.

Comment: The introduction section has been redrafted (rows 47-91), focusing more on highlighting in a clear manner the epistemological structure, the novelty of the study, the research aim, and the added value in the field. Moreover, the literature review section has been restructured, following a more systematic approach, and focusing more on emphasizing the research gap. 

2. The two-stages of the empirical analysis are meaningful and complement each other. The description of the methodological features of both methods is concise, but orients the reader especially if he is already familiarized with these methods. However, I feel that the main drawback of this analysis lies in the variables selection process. 

In both cases (input and output variables for DEA, respectively explanatory variables for the quartile regression) there is no convincing substantiation of the choice of these particular variables. Why these and not others? What is their informational content, their theoretical/intuitive relationship with the dependent variables? For example, the choice of input and output variables included in the DEA model specification is highly influencing the computation of the efficiency scores. Any variable change may significantly alter the scores and hence the efficiency status of a country. This part of the study needs to be better explained, in order to support the reliability and soundness of the findings.

Comment: The selection of input and output variables, as well as explanatory variables for the regression part of the methodology, is now argued in much more detail in the manuscript, based on and in relation to the relevant literature. 

The section that sustains the choice of variables has been moved from the literature section to the data and methodology section, and it was further completed.

Selection of input variable: rows 242-252

Selection of output variables: rows 255 - 276

Selection of explanatory variables: rows 296-387

3. The figures in the Annexes have no title.

Comment: All additional files have now been carefully formatted and named to meet PLOS ONE’s style requirements. 

Best regards, 

The authors

---

## [Editor Report · Decision Letter 1]

22 Aug 2023

Public sector’s efficiency as a reflection of governance quality, an European Union study

PONE-D-23-15310R1

Dear Dr. Dinca,

We’re pleased to inform you that your manuscript has been judged scientifically suitable for publication and will be formally accepted for publication once it meets all outstanding technical requirements.

Kind regards,

Javier Cifuentes-Faura

Academic Editor

PLOS ONE

Additional Editor Comments (optional):

Congratulations
---

## [Editor Report · Acceptance letter]

29 Aug 2023

PONE-D-23-15310R1 

Public sector’s efficiency as a reflection of governance quality, an European Union study 

Dear Dr. Dinca:

I'm pleased to inform you that your manuscript has been deemed suitable for publication in PLOS ONE. Congratulations! Your manuscript is now with our production department. 

Kind regards, 

on behalf of

Prof Javier Cifuentes-Faura 

Academic Editor

PLOS ONE